# ARE HALLUCINATIONS BAD ESTIMATIONS?

## ABSTRACT

We formalize hallucinations in generative models as failures to link an estimate to any plausible cause. Under this interpretation, we show that even loss-minimizing optimal estimators still hallucinate. We confirm this with a general high probability lower bound on hallucinate rate for generic data distributions. This reframes hallucination as structural misalignment between loss minimization and human-acceptable outputs, and hence estimation errors induced by miscalibration. Experiments on coin aggregation, open-ended QA, and text-to-image support our theory.

## 1 INTRODUCTION

*Hallucination* in generative model refers to a model generating confident yet unsupported or non-factual outputs. This failure undermines user trust, safety, and the practical utility of AI systems. It becomes a critical concern in modern machine learning with the widespread deployment of large-scale generative models across language, vision, and multimodal domains (Ji et al., 2023; Liu et al., 2024; Bai et al., 2024; Kalai et al., 2025). To address it, we must understand why models hallucinate at a fundamental level. In this work, we formalize hallucination as an attribution failure: the *estimated prediction* does not align with any *plausible input cause* under standard loss-minimizing training. From this perspective, we prove hallucination persists even for Bayes-optimal estimators.

Prior theory attributes hallucination to resource limits, sparse data, or computational hardness. Xu et al. (2024) study hallucination as the mismatch between a model's computed function and the ground-truth function. They prove that any polynomial-time language model hallucinates on some tasks due to computational limits. Kalai & Vempala (2024) show that even a calibrated model hallucinates on rare "singleton" facts. They lower bound the hallucination rate by the frequency (redundancy) of these facts in the training data. Banerjee et al. (2024) study hallucination through Gödel's first incompleteness theorem. They argue that no finite dataset captures all valid inferences, so hallucination persists regardless of model or data scale. Taken together, these results frame hallucination as a byproduct of constraints rather than a structural feature of estimation.

In contrast, we posit that hallucination is not only a symptom of modeling limitations but also a structural phenomenon of estimation itself. Our key insight is that hallucinations may still persist even for Bayes-optimal estimators with unlimited capacity that minimize the true training loss. In other words, a model with infinite power, trained without resource constraints, still outputs implausible content. The crux is a misalignment between the model's objective and human expectations. A loss-minimizing model is optimized to produce the average outcome, whereas a human evaluator expects a specific plausible outcome (typically, one of the modes of the true distribution).

This reframes hallucination as *structural misalignment*. Hallucination is a manifestation of estimation errors induced by miscalibration. To be concrete, under expected standard loss, the Bayes-optimal predictor for a target distribution $A(X)$ given the input $X$ is the conditional expectation

$$A^*(X) = \mathbb{E}[A(X)],$$

which minimizes the expected error by construction. If the true conditional distribution $\Pr[A(X)] = \Pr[A(x) \mid X = x]$ is multimodel[1], then $A^\star(X)$ average across all those possible outcomes and may fall in a low-probability region. It matches none of the plausible modes. The estimate minimizes error yet fails to align with any realistic ground-truth outcome. Thus even an optimal estimator may produce outputs that no human would recognize as valid or plausible. We deem this is a fundamental source of hallucination in generative models. To this end, we formalize this into $\delta$-*hallucination*: an estimator's

---

[1]For instance, an open-ended question that has several distinct correct answers.

output that lies outside a $\delta$-neighborhood of every plausible outcome (please see Section 3 for precise definitions.) This reframing shows hallucination as a consequence of the objective misalignment, rather than just a lack of model capacity or data.

**Contributions.** Our contributions are as follows.

- **New Formulation for Hallucination Fundamental Source.** We characterize hallucination phenomena in generative models by introducing $\delta$-hallucination. This interprets hallucination as outputs that fail to match any plausible human-acceptable outcome. The formulation provides a rigorous and measurable way to analyze hallucination in generative models.

- **Hallucination of Optimal Estimators.** We prove that loss-minimizing optimal estimators still $\delta$-hallucination. We extend the result to near-optimal estimators, to multiple inputs, and to inputs with hinted latent variables. These results confirm hallucination as a fundamental source rooted in the estimation process itself.

- **Fundamental Limits of Hallucination.** We derive a general lower bound on the probability of $\delta$-hallucination under mild distribution assumptions. This bound reaffirms that hallucinations persist at a non-zero rate. This establishes a fundamental limit that prevents eliminating the source of hallucinations through larger models or datasets.

- **Experiment Validation.** We validate our theory through controlled experiments on coin-flipping aggregation, open-ended QA, and text-to-image generation. The results demonstrate that minimizing loss does not remove hallucination. The persistence across both synthetic and real-world settings confirms hallucination as a structural feature of estimation and a fundamental source of model misalignment.

**Organization.** Section 3 defines hallucination as $\delta$-hallucination. Section 4 demonstrates hallucination of optimal estimators. Section 5 provides a lower bound on the probability of hallucination. Section 6 details experiment results.

**Related Work.** We defer related work discusssion to Appendix A due to page limits.

## 2 PRELIMINARIES

**Notations.** In this work, $f_Y(\cdot)$ denotes the probability density function over the randomness of $Y$. $\mathbb{E}_Y[T]$ denotes the expectation of a random variable $T$ over $Y$. $[N]$ denotes the set: $\{1, 2, \cdots, N\}$. $\|\cdot\|_2$ denotes 2-norm. We use $\|\cdot\|_2$ as the square root of the square sum of all entries. For a column vector $v$, we use $v_i$ to denote its $i$-th entry from the top. For a matrix $M$, we use $M_{r,c}$ to denote its entry at $r$-th row and $c$-th column. We write $M_{:,c}$ and $M_{r,:}$ to denote its $c$-th column and $r$-th row, respectively. We use $1_a$ to denote an indicator that is 1 when $a$ happens and 0 otherwise.

**Expected Quadratic Loss.** We define expected quadratic loss as follows.

**Definition 2.1** (Expected Quadratic Loss). Let $X$ be an input, let $A(X)$ be a random target output associated with $X$, and let $A^*(X)$ be an estimator for $A(X)$. Define the expected quadratic loss of the estimator $A^*(X)$ with respect to the true output $A(X)$ as:

$$\ell_A(A^*(X)) := \mathbb{E}\left[\|A^*(X) - A(X)\|_2^2\right].$$

In other words, $\ell_A(A^*(X))$ is the expected squared $\ell_2$ error between the estimate and the actual outcome. This quantity serves as the objective that an optimal estimator would minimize (e.g., the Bayes-optimal estimator minimizes the expected quadratic loss by construction).

**Remark 2.1.** We use the $\ell_2$ loss in the main text for clarity of exposition. In Appendix D, we show that all results remain valid under the cross-entropy loss, which is the standard training objective for generative models in self-supervised learning. This extension is natural because cross-entropy is a *proper scoring rule*: its Bayes-optimal solution is the true conditional distribution $P(Y|X)$, so the same structural arguments for $\delta$-hallucination continue to apply.

We use the expected quadratic loss to formalize the objective minimized by an optimal estimator.

**Lipschitzness.** We define Lipschitzness in 2-norm as follows.

**Definition 2.2** (Lipschitzness). We say a function $g$ is $L$-Lipschitz (with respect to the $\ell_2$-norm) if there exists a constant $L > 0$ such that for all inputs $x$ and $y$ in its domain

$$\|g(x) - g(y)\|_2 \leq L\|x - y\|_2.$$

We use Lipschitzness to impose a regularity condition on the estimator. This condition ensures that small changes in the input lead to at most $L$-scaled changes in the output. In our analyses, we assume Lipschitzness as a smoothness property that rules out estimators with abrupt or unstable behavior.

**Latent Variable $Z$.** In the context of self-supervised learning, we represent the output of the model as a probability distribution (Devlin et al., 2019; Radford et al., 2021). Specifically, when an estimator outputs contextual factors such as speaker attitude or intended audience, we may categorize the possible outputs based on the specific factors they exhibit. Then, we see different categories (which are sub-distributions in the original target distribution) as conditional distributions under different states of a latent variable $Z$. We illustrate the concept of this latent variable $Z$ in Figure 1.

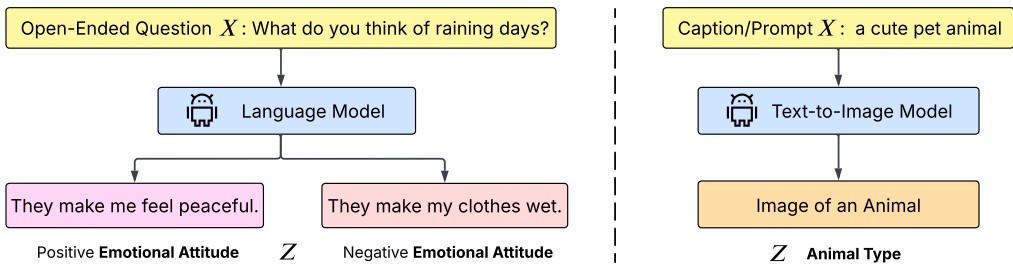

Figure 1: **Examples of Latent Variable $Z$.** For an open-ended question or prompt $X$, the latent variable $Z$ may be the emotional attitude or categories in the target distribution.

## 3 $\delta$-HALLUCINATION

We present our definition of $\delta$-hallucination as the gap between objective optimized by the model and the underlying causes of variation ($Z$). That is, conditioning on the state of $Z$ changes the distribution of the output. We begin by defining the relation between input $X$ and latent variable $Z$ as follows.

**Definition 3.1** (Data Distribution and Latent Variable). Let $X \in \mathbb{R}^{d_x}$ denote the input, and let $A(X) \in \mathbb{R}^{d_a}$ denote a random variable representing the target output associated with $X$, where $d_x$ and $d_a$ are the input and output dimensions. Let $Z$ be a latent variable associated with $X$, and let $\{Z_i\}_{i\in[N]}$ denote its possible states. The conditional output random variable given $Z_i$ is

$$A(X; Z_i) := A(X) \mid \{Z = Z_i\},$$

which represents the target output distribution of $X$ under latent state $Z_i$. If probability densities exist, the conditional density is

$$f_{A(X;Z_i)}(a) := \frac{f_{A(X),Z}(a, Z_i)}{\Pr[Z = Z_i]},$$

where $f_{A(X),Z}$ is the joint density of $(A(X), Z)$.

**Remark 3.1.** $A(X)$ in Definition 3.1 defines the data distribution, but we also view it as the real distribution in this paper. Intuitively, $Z$ indexes hidden causes that resolve ambiguity in the output. $A(X; Z_i)$ isolates the distribution of valid outputs when the hidden cause equals $Z_i$. The marginal $A(X)$ mixes these conditional laws with weights $\Pr[Z = Z_i]$, so multi-modality in $A(X)$ arises from variation over $Z$.

**Key Insight.** While minimizing the loss on the whole data distribution is critical for model estimations, it is *also important to*

$$\max_{i \in [N]} \{ f_{A(X;Z_i)}(A^*(X)) \},$$

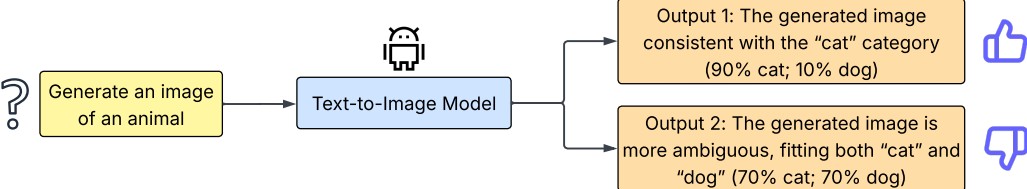

Figure 2: **An Example of Our Key Insight.** Suppose the open-ended question is to generate a picture of an animal. Then the output with 90% of conditional probability under the category of cat and a 10% of conditional probability under the category of dogs is considered better than the output which has a 70% of probability density under the category of cat and 70% under the category of dog.

which is the maximum probability density of the estimate $A^*(X)$ under $Z = Z_i$. This reflects that a good estimate aligns with at least one plausible underlying state rather than consistent with all. We give an example in Figure 2 to illustrate the interpretation.

Formally, we present the above insight as $\delta$-hallucination.

**Definition 3.2** ($\delta$-Hallucination). Let $X$ be an input and $Z$ a latent variable associated with $X$ taking values in $\{Z_i\}_{i \in [N]}$. Fix a tolerance parameter $\delta \in (0, 1]$, and let $A^*$ be an estimator of $X$. We say that $A^*$ $\delta$-*hallucinates* at $X$ if, for every $i \in [N]$,

$$f(A(X; Z_i) = A^*(X)) \leq \delta, \quad i \in [N],$$

where $f_{A(X;Z_i)}$ denotes the probability mass function (in the discrete case) or probability density function (in the continuous case) of $A(X; Z_i)$.

That is, for every possible latent state, the probability of producing the estimated output $A^*(X)$ does not exceed $\delta$. In other words, Definition 3.2 implies that $\delta$-hallucination is a generated answer that has low calculated loss but is unlikely to belong to any state or class of possible outputs.

**Remark 3.2.** Intuitively, $\delta$-hallucination occurs when the estimator $A^*(X)$ outputs a value that has low likelihood under *every* plausible latent state of $Z$. In such a case, the prediction fail to be attributed to any genuine cause consistent with the data distribution. This captures the idea that hallucination arises not merely from error, but from producing an output that fails to align with any valid mode of the underlying conditional distributions.

## 4 OPTIMAL ESTIMATOR STILL HALLUCINATES

We establish the existence of $\delta$-hallucination. We begin with the single-input case, showing that even an optimal estimator minimizing loss may $\delta$-hallucinate, and that this extends to semi-optimal estimators within $\epsilon$ of the optimum. We then extend the result to the multi-input setting. Finally, we consider the practical case where the model receives hints about hidden influences in the input, and show that hallucination exists under standard regularity conditions.

$\delta$**-Hallucination Under a Single Input.** We show that even an loss-minimizing optimal estimator may output an answer that $\delta$-hallucinates by Definition 3.2.

**Theorem 4.1** (Existence of $\delta$-Hallucination Under Single Input). For an input $X$, there exists infinitely many distributions of $A(X)$ and $Z$ such that for an estimator $A^*$ that minimizes the expected quadratic loss defined in Definition 2.1 over $A(X)$, it is bound to $\delta$-hallucinate at $X$.

*Proof.* See Appendix C.1 for detailed proof. □

We further demonstrate the existence of $\delta$-hallucination on semi-optimal estimators.

**Theorem 4.2** (Existence of $\delta$-Hallucination on Semi-Optimal Estimators under Single Input). For an input $X$, there exists infinitely many distributions of $A(X)$ and $Z$ such that if an estimator $A'$ is within a distance of $\epsilon$ to the optimal estimator $A^*$, which writes as

$$\|A'(X) - A^*(X)\|_2 \leq \epsilon,$$

then $A'(X)$ is bound to $\delta$-hallucinate.

*Proof.* See Appendix C.3 for detailed proof. □

**$\delta$-Hallucination under Multiple Inputs.** When considering a collection of inputs, our definition applies to each input individually. We describe the $\delta$-hallucination under multiple inputs as follows.

**Corollary 4.2.1** (Existence of $\delta$-Hallucination under Multiple Inputs). For a set of input $X_j, j \in [S]$, there exists infinitely many distributions of $A(X_j)$ and $Z$ such that any estimator minimizing the expected quadratic loss defined in Definition 2.1 is bound to $\delta$-hallucinate at $X$.

*Proof.* See Corollary C.1.1 for detailed proof. □

**$\delta$-Hallucination with Hinted Latent Variables.** In practical situations, the model receives hints about hidden influences in the input. We define this hint as a tilt upon the input $X$ as follows.

**Definition 4.1** (Effect of Latent Variable on Input). For an input $X$, let $A(X)$ be its target distribution. For a latent variable $Z$ associated with $X$, let $Z_i$ denote the states of this latent variable, and let $\delta_i$ denote a hint for the state $Z_i$ for all $i \in [N]$, which satisfies

$$A(X + \delta_i) = A(X; Z = Z_i), \quad i \in [N].$$

This means the target distribution of the tilted input is the posterior distribution when knowing $Z = Z_i$.

Based on Definition 4.1, we show $\delta$-hallucination exists for tilted input under Lipschitzness regularity condition as follows.

**Theorem 4.3** (Existence of $\delta$-Hallucination at Tilted Input). Let $B_\delta$ denote the bound of all hints $\delta_i, i \in [N]$, defined as

$$B_\delta := \sup_{i \in [N]} \|\delta_i\|_2.$$

For an $L$-Lipschitz estimator $A^*$ satisfying Definition 2.2, there exists infinitely many distributions of $A(X; Z)$ such that $\delta$-Hallucination happens on all $X + \delta_i$. That is, $A^*(X + \delta_i)$ does not fall into the region where $f_{A(X;Z_i)} \geq \delta$ for any $i \in [N]$ by Definition 3.1.

*Proof.* See Appendix C.4 for detailed proof. □

Thus, we show that hallucination is intrinsic to the probabilistic structure of estimation, across optimal and near-optimal estimators, multiple inputs, and even when the answers' directions are hinted.

## 5 HALLUCINATION PROBABILITY LOWER BOUND

We extend our result beyond existence of $\delta$-hallucination in Section 5 and provide a lower bound on the probability of hallucination for optimal estimators satisfying certain conditions.

We begin with the definition of means and variances for the variables of interest.

**Definition 5.1** (Means and Variances). Let $\{Z_i\}_{i \in [N]}$ denote the possible states of the latent variable $Z$, with probabilities $p_i := \Pr[Z = Z_i]$. For each $i \in [N]$, define the conditional mean

$$\mu_i := \mathbb{E}[A(X; Z_i)].$$

We regard $\mu_i$ as a realization of a random variable distributed according to $d_i^\mu$. Let $\mu_i^d := \mathbb{E}_{d_i^\mu}[\mu_i]$ and $\sigma_i^d := \mathrm{Var}_{d_i^\mu}[\mu_i]$ denote the mean and variance of this distribution, respectively. Let $d^\mu$ denote the joint distribution of $(\mu_1, \ldots, \mu_N)$. We write $\mu^d := \mathbb{E}_{d^\mu}[\mu_1, \ldots, \mu_N]$ for its mean vector and $\sigma^d := \mathbb{E}[\sum_{i=1}^N (\mu_i - \mu_i^d)^2]$ as sum of variance.

We then provide the following assumptions applied to $\mu_i$ and $d_i^\mu$ in Definition 5.1. In particular, we assume that the conditional means align around a common value and that the joint distributions of these conditional means are mutually independent.

**Assumption 5.1.** We impose the following conditions on the distributions defined in Definition 5.1:

1. *Identical means*: There exists a constant $\mu_0 \in \mathbb{R}$ such that $\mu_i^d = \mu_0$, for all $i \in [N]$.
2. *Independence*: The distributions $\{d_i^\mu\}_{i=1}^N$ are mutually independent.

We now characterize hallucination events in terms of output regions that correspond to high ($> \delta$) conditional probability under each latent state.

**Definition 5.2** (High Conditional Density Regions). We define $U_i^\delta$ to be

$$U_i^\delta := \{a \mid f(a; Z_i) > \delta\},$$

which is the region with posterior probability of $Z = Z_i$ larger than $\delta$.

**Remark 5.1.** By Definition 5.2, $\delta$-hallucination of $A^*(X)$ is equivalent to

$$A^*(X) \notin U_i^\delta, \quad i \in [N].$$

**Remark 5.2.** We highlight the relationship between Highest Conditional Density Regions (HCDRs) and the classical Highest Density Regions (HDRs) (Caprio et al., 2024; Dahl et al., 2024). When the latent variable $Z$ has only a *single* state, $\delta$-hallucination reduces to the event that the target distribution falls outside the HDR of a given mass, where the mass corresponds to a density threshold $\delta$. When $Z$ has *multiple* states, we generalize this idea by introducing HCDRs, which capture high-density regions conditioned on each latent state. See Appendix B for definitions and a detailed discussion.

We then define the following spheres covering $U_i^\delta$ in Definition 5.2. Specifically, we enclose each $U_i^\delta$ within the smallest possible sphere centered at the corresponding mean $\mu_i$.

**Definition 5.3** (Minimal Covering Spheres). For each $i \in [N]$, let $U_i^\delta \subset \mathbb{R}^{d_a}$ denote the $\delta$-high density region associated with state $Z_i$. Define $B_i^\delta(r)$ as the closed Euclidean ball of radius $r$ centered at $\mu_i$. The minimal covering radius is

$$r_i := \inf_{r_i \in \mathbb{R}^+} \{U_i^\delta \subset B_i^\delta(r_i)\}.$$

Thus $B_i^\delta(r_i)$ is the smallest sphere centered at $\mu_i$ that contains $U_i^\delta$. Finally, define the uniform covering radius

$$r = \max_{i \in [N]} \{r_i\}.$$

**Remark 5.3.** Geometrically, $r_i$ measures the worst-case deviation of the $\delta$-high density region $U_i^\delta$ from its center $\mu_i$. In other words, it is the maximum distance one must travel from $\mu_i$ to reach any point in $U_i^\delta$. The uniform covering radius $r$ then gives a single bound that applies across all latent states, capturing the largest such deviation. This interpretation is useful for intuition: $r_i$ quantifies how "spread out" the high-density region is around its mean, while $r$ aggregates the largest of these spreads across all $i$.

With definitions and assumptions established, we now derive a lower bound on the probability of hallucination for any optimal estimator.

**Theorem 5.1** (Hallucination Probability Lower Bound). Let $(A(X), Z)$ satisfy Assumption 5.1. For each $i \in [N]$, let $\mu_i, \sigma_i^d$ be as in Definition 5.1, let $\mu_0$ be as in Assumption 5.1, and let $r_x$ be as in Definition 5.3. Define

$$d := (\sum_{j=1}^N p_j^2 \sigma_j^d)^{1/2}, \quad \theta_i(\alpha) := \frac{(\alpha d + r_x)^2}{\sigma_i^d}, \quad \alpha > 1, \quad \text{and} \quad K_i^\mu := \frac{(\mathbb{E}[(\mu_i - \mu_0)^2])^2}{\mathbb{E}[(\mu_i - \mu_0)^4]}.$$

If for every $i \in [N]$ there exists $\alpha_i > 1$ such that $\theta_i(\alpha_i) \leq 1$, then

$$P_H^\delta > \prod_{i=1}^N (P_i K_i^\mu),$$

where $P_H^\delta$ denotes the probability that the optimal estimator $A^*$ $\delta$-hallucinates at $X$ (equivalently, $A^*(X) \notin U_i^\delta$ for all $i \in [N]$, with $U_i^\delta$ as in Definition 5.3).

*Proof.* See Appendix C.5 for detailed proof. □

## 6 EXPERIMENTS

We validate our interpretations and claims with three complementary experiments. In particular, we first provide a synthetic coin-flipping problem (Section 6.1) where it demonstrates that models trained purely with likelihood objectives shows persistent $\delta$-hallucination. We then extend these insights to large-scale LLM (Section 6.2) and text-to-image generation (Section 6.2) settings. Both experiments validate our claim that a loss-minimizing optimal estimator $\delta$-hallucinates.

### 6.1 SYNTHETIC COIN FLIPPING PROBLEM

**Objective.** We evaluate our claim that minimizing loss may not increase the conditional probability of estimated output with respect to input labels as in Theorem 4.1.

**Experiment Design.** We design a controlled experiment based on the classical coin-flipping problem. We choose a subset of coins from a collection of coins (each with a distinct probability of landing heads), flip them, and record the total number of heads observed. The model receives the labels of the chosen coins as input. We then train the model to predict the recorded total. These labels do not explicitly reveal the head probabilities, and thus act as latent hints rather than explicit supervision.

**Data.** We generate $2N$ coins, each with a unique head probability, and perform $M$ flips to construct the dataset. We consider $N = 2, 3$, and $5$, with $M$ ranging from $20000$ to $40000$.

**Model Architecture.** We adopt an 8-layer transformer with 64 hidden dimensions and 256 feed-forward dimensions for this experiment.

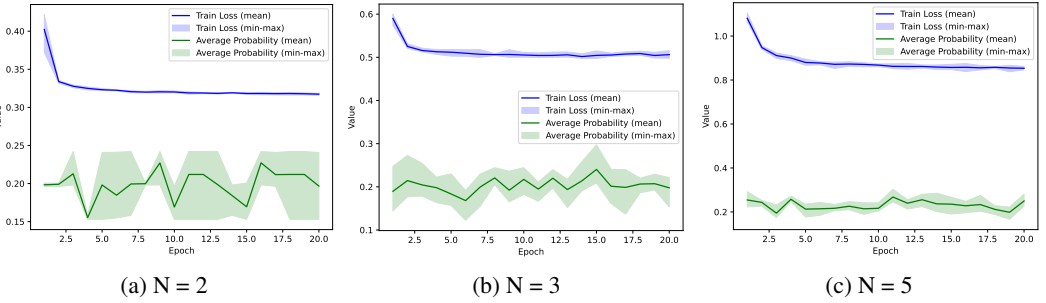

|              (a) N = 2              |              (b) N = 3              |              (c) N = 5              |

Figure 3: We conducted 5 rounds of experiments on each of $N = 2, 3$ and $5$. The results show that training loss does not correlate with the conditional probability of the model estimation with respect to input labels. This aligns with our theoretical result in Theorem 4.1.

**Results.** As shown in Figure 3, we observe that the descent of training losses does not correlate with the rise or drop of the conditional probability of the estimations generated on the validation set. This result aligns with our theoretical claim that minimizing the loss does not necessarily maximize the conditional probability (of a latent state) of the estimate.

### 6.2 OPEN-ENDED TEXT QUESTIONS

**Objective.** We evaluate hallucination in the LLM models by measruing the resemblance of model output to the commonly incorrect answers in TruthfulQA (Lin et al., 2021).

**Experiment Design.** We fine-tune pretrained language models on a dataset of open-ended questions and compare their outputs to those of

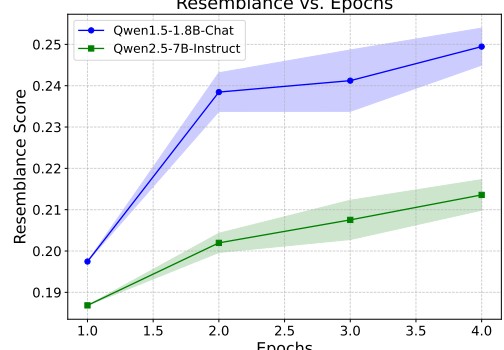

Figure 4: **Resemblance vs. Epochs.** We fine-tune Qwen1.5-1.8B-Chat and Qwen2.5-7B-Instruct for 2, 3, and 4 epochs and test the answers' resemblance to commonly incorrect answers in TruthfulQA. We repeat this process for 2 random seeds. Results validate that hallucination persists even as the model minimizes its predictive objective.

the original models. We measure the the model's tendency to resemble the commonly incorrect answers in TruthfulQA (Lin et al., 2021). We use Gestalt Pattern Matching (difflib in Python) to measure resemblance.

**Data.** We use GPT5, Gemini 2.5 Flash, and DeepSeek R1 to generate a dataset of 300 open-ended questions with 2 possible answers. This forms a dataset of 600 question-answer pairs.

**Model Architecture.** We fine-tune Qwen1.5-1.8B-Chat and Qwen2.5-7B-Instruct on our open-ended question dataset using LLaMA-Factory with LoRA adapters.

**Results.** As shown in Figure 4 and Table 1, both models show a consistent increase in resemblance over additional fine-tuning epochs. The results reveal that, though we fine-tune the models to obtain low predictive loss, both models become more aligned with commonly incorrect answers. This pattern is consistent across all seeds as shown in Table 1. The finding supports our theoretical claim that loss minimization alone is insufficient to eliminate $\delta$-hallucination.

Table 1: **Resemblance of Fine-Tuned Models' Answers to Commonly Incorrect Answers in TruthfulQA**. Each model is fine-tuned for 2, 3, and 4 epochs with 2 random seeds. The resemblance does not decrease with training, validating that hallucination persists in loss-minimizing optimal models.

| Epochs | Qwen1.5-1.8B-Chat | | Qwen2.5-7B-Instruct | |
|---|---|---|---|---|
| | Seed 1 | Seed 2 | Seed 1 | Seed 2 |
| Original | 0.1975 | – | 0.1868 | – |
| 2 | 0.2338 | 0.2431 | 0.2043 | 0.1997 |
| 3 | 0.2338 | 0.2486 | 0.2123 | 0.2028 |
| 4 | 0.2450 | 0.2539 | 0.2173 | 0.2099 |

## 6.3 OPEN-ENDED TEXT-TO-IMAGE

**Objective.** We evaluate hallucination in a text-to-image setting where we detect generated samples falling outside a calibrated HCDR as in Definition C.2 and Remark 5.2.

**Experiment Design.** We first construct HCDR from real AFHQ cat and dog images. We begin by extracting fixed CLIP embeddings from the images, which are then normalized, reduced in dimension via PCA, and standardized through z-scoring. For each class (cats, dogs), we fit a Gaussian Mixture Model (GMM) on an 80% training split of the preprocessed embeddings to learn what cat or dog features look like. We then use the remaining 20% testing data to obtain log-densities and compute a class-specific threshold at the 10% percentile. This threshold corresponds to a cutoff such that the top 90% of the testing images are included in the HDR for each class (See Figure 7 of Appendix B for a visualization of HDR for cats and dogs). In other words, a new embedding is considered to lie outside of HDR or a specific class if its log-likelihood under that class's GMM exceeds the threshold. Finally, to form HCDR, we take the union of the per-class HDRs: a generated embedding is inside the HCDR if it lies in at least one class HDR, and outside otherwise.

We then fine-tune a text-to-image generative model, with the text encoder frozen, on the training dataset for the model to mainly learn the image distribution (target). We evaluate the portion of generated images outside of HCDR for given prompts.

**Data.** We use Animal Faces-HQ (AFHQ) (Choi et al., 2020). We extract 5558 cat images and 5139 dog images. Each is 512 by 512 pixels. We construct 3 prompts for evaluation: "a realistic photo of a friendly dog", "a fluffy cat sitting on a sofa", and "a cute pet animal".

**Model Architecture.** We use CLIP ViT-B/32 model model to extract image CLIP embeddings. For generation, we fine-tune the UNet component of Stable Diffusion v1.5, while keeping the text encoder and VAE frozen.

**Results.** As shown in Figure 5, as we fine-tune the model, the training loss decreases, indicating that the model captures the distribution of the dataset, yet hallucination rate do not converge. It supports our theoretical claim that loss minimization alone is insufficient to eliminate $\delta$-hallucination.

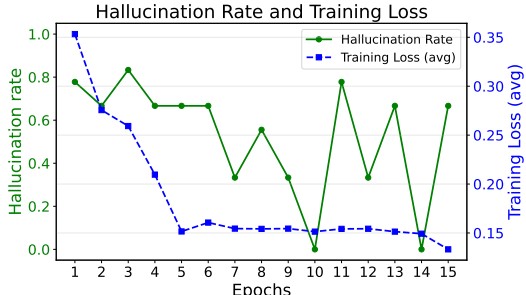

Figure 5: **Hallucination Rate and Training Loss.** We plot hallucination rate (green, left axis) and training loss (blue, right axis) over epochs. While the training loss decreases, the hallucination rate does not converge and often fluctuates, showing that hallucination persists even as the model minimizes its predictive objective.

### 6.3.1 ABLATION STUDY ON PROMPTS

We further conduct studies on 3 types of prompts for the text-to-image generative model: one targeting the cat category ("a fluffy cat sitting on a soft"), one targeting the dog category ("a realistic photo of a friendly dog"), and one mixed prompt ("a cute pet animal"). We evaluate the hallucination rate for each prompt across training epochs. As shown in Figure 6, our results consistently show that, even under a loss-minimizing estimator, hallucinations persist and do not converge to zero. This indicates that even when prompts hint information about target category, hallucinations may still occur.

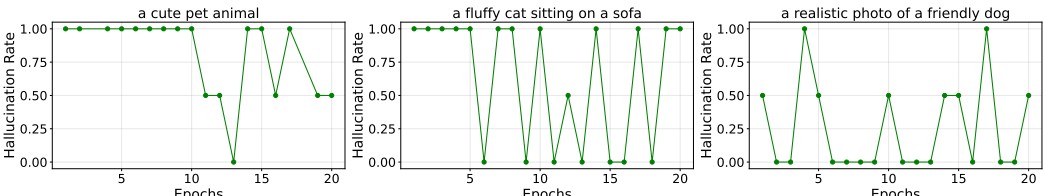

Figure 6: **Prompts Analysis.** We create 3 types of prompts and evaluate their hallucination rate respectively. All plots show even a loss-minimizing estimator hallucinates.

## 7 CONCLUSION

In this work, we reframed hallucination in generative models as a fundamental misalignment between standard loss-based training objectives and human expectations. Under this view, we formalized $\delta$-hallucination to capture when an estimator's output fails to match any plausible real-world outcome (Section 3). Crucially, we showed that no amount of model capacity or data can eliminate hallucinations: even an ideal Bayes-optimal estimator (one minimizing the true expected loss) may still generate implausible predictions on inputs with inherently diverse correct answers (Section 4). We derived general lower bounds on how frequently such hallucinations must occur for broad classes of target distributions (Section 5), and validated these predictions with both synthetic and real-world experiments (Section 5). Taken together, our findings establish that hallucination is a structural property of the estimation process itself rather than just a symptom of limited models or datasets.

**Limitations.** While our theory offers a new perspective on hallucinations, it has a few limitations. The current lower bound for $\delta$-hallucination is relatively loose and relies on certain assumptions, leaving room for tighter bounds under more relaxed conditions. Additionally, our analysis focused on a general estimator. Examining specific model families or tasks might yield stronger guarantees or further insight into when and how hallucinations arise.

**Implications and Future Work.** By identifying hallucination as arising from the core training objective, our results imply that simply scaling up model size or dataset coverage is insufficient to eliminate the problem. Effective mitigation may require rethinking generative model training, with objectives explicitly aligned to human standards of correctness. In practice, this could mean favoring more *mode-seeking* behavior —generating high-probability, consistent outputs — rather than minimizing average error across all possible outcomes. Future training methods may need to incorporate constraints or decision-theoretic criteria that push models to commit to a single plausible answer instead of blending incompatible modes. Several concrete directions follow from our findings:

- **Alternative Loss Functions.** Extend our theoretical framework to other loss functions to investigate how the choice of training objective influences hallucination rates.
- **Alignment-Oriented Training Schemes.** Design practical strategies that scale our insights, such as HDR-guided sampling or mixed-objective fine-tuning that explicitly penalizes implausible outputs.
- **Multimodal and Structured Outputs.** Generalize the analysis to multimodal and structured tasks, where the space of valid outputs is richer, to uncover new alignment strategies tailored to complex domains.

In summary, treating hallucination as a structural phenomenon calls for a shift away from naive average-case error minimization and toward objectives that explicitly prefer outputs aligned with one of the true modes, thereby better matching human standards of reliability.

ETHIC STATEMENT

This paper does not involve human subjects, personally identifiable data, or sensitive applications. We do not foresee direct ethical risks. We follow the ICLR Code of Ethics and affirm that all aspects of this research comply with the principles of fairness, transparency, and integrity.

REPRODUCIBILITY STATEMENT

We ensure reproducibility of our theoretical results by including all formal assumptions, definitions, and complete proofs in the appendix. The main text states each theorem clearly and refers to the detailed proofs. For experiments, we describe model architectures, datasets, preprocessing steps, hyperparameters, and training details in the main text. Code and scripts are provided in the supplementary materials to replicate the empirical results.

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

# Appendix

## IMPACT STATEMENT

By the theoretical nature of this work, we do not anticipate any negative social impact.

## LLM USAGE DISCLOSURE

We used large language models (LLMs) to aid and polish writing, such as improving clarity, grammar, and conciseness. We also used LLMs for retrieval and discovery, for example exhausting literature to identify potential missing related work. All technical content, proofs, experiments, and results are original contributions by the authors.

## A  RELATED WORKS

Hallucinations in generative models have been studied from both theoretical and empirical perspectives. Prior theory frames them as inevitable outcomes of practical limits: finite parameters, sparse data, or computational hardness. (Xu et al., 2024) prove that any polynomial-time language model hallucinates on certain tasks. Kalai & Vempala (2024) show that even a calibrated model hallucinates at a rate tied to the fraction of "singleton" facts that appear only once in the training set. Banerjee et al. (2024) argue that no finite dataset or architecture covers all valid inferences, ensuring a nonzero hallucination rate regardless of scale. These works treat hallucination not as a flaw in estimation itself, but as an artifact of underfitting caused by resource and computational limits. More recently, Kalai et al. (2025) propose that hallucination stems from mismatches between predictive likelihood training, incomplete coverage, and reinforcement learning, suggesting hallucinations persist even with scale and motivating deeper foundational study.

Recent empirical research has delivered taxonomies, benchmarks, and mitigation techniques for hallucinations in generative models. Huang et al. (2025) survey intrinsic and extrinsic hallucinations, and review detection and mitigation methods. Ji et al. (2023) provide a broad overview of metrics and task-specific phenomena across summarization, dialogue, and machine translation. Zhang et al. (2023) analyze detection and explanation methods. Li et al. (2024) conduct a factuality study, introducing a new benchmark and evaluating detection, sources, and mitigation. Farquhar et al. (2024) propose entropy-based uncertainty estimators to detect confabulations. In contrast to viewing hallucinations only as limitations, Jiang et al. (2024) explore their creative potential. A notable work by Aithal et al. (2024) analyzes hallucinations in diffusion models and attributes them to mode interpolation, where samples fall into regions not supported by training data. Their empirical observations support our theoretical findings by linking artifacts beyond data support to interpolation between nearby modes (corresponding to regions with low conditional probability density under any latent state in our work).

Building on prior work, we propose a new interpretation of hallucination: it arises from a gap between model training objectives and human criteria. Estimation fails when outputs do not align with any plausible human-perceptive category. We formalize this gap as $\delta$-hallucination and prove that even loss-minimizing optimal estimators produce outputs with low conditional probability under every category. We derive a general lower bound on the probability of $\delta$-hallucination and validate

our claims with empirical studies. These results establish hallucination as a structural feature of estimation itself, not a flaw of model size, data coverage, or specific queries.

## B HIGHEST CONDITIONAL DENSITY REGIONS

**Highest Density Regions.** (Hyndman, 1996) popularize the concept of Highest Density Regions (HDRs) as the smallest-volume set containing a given probability mass He provided practical algorithms for computing and visualizing HDRs for univariate and multivariate densities, showing their advantages over equal-tailed intervals in revealing multi-modal structure. (Samworth & Wand, 2010) developed a rigorous asymptotic theory for kernel-based HDR estimation, deriving uniform-in-bandwidth risk approximations and proposing optimal bandwidth selectors that minimize HDR estimation error. (Haselsteiner et al., 2017) introduced the idea of using HDRs to define environmental-contours—termed highest-density contours—in engineering design, demonstrating that HDR-based contours yield more compact, interpretable regions for multimodal environmental distributions.

In a concrete example, we build calibration datasets for the categories of cats and dogs in AFHQ dataset (Choi et al., 2020) and estimate their log-densities under GMM model as shown in Figure 7.

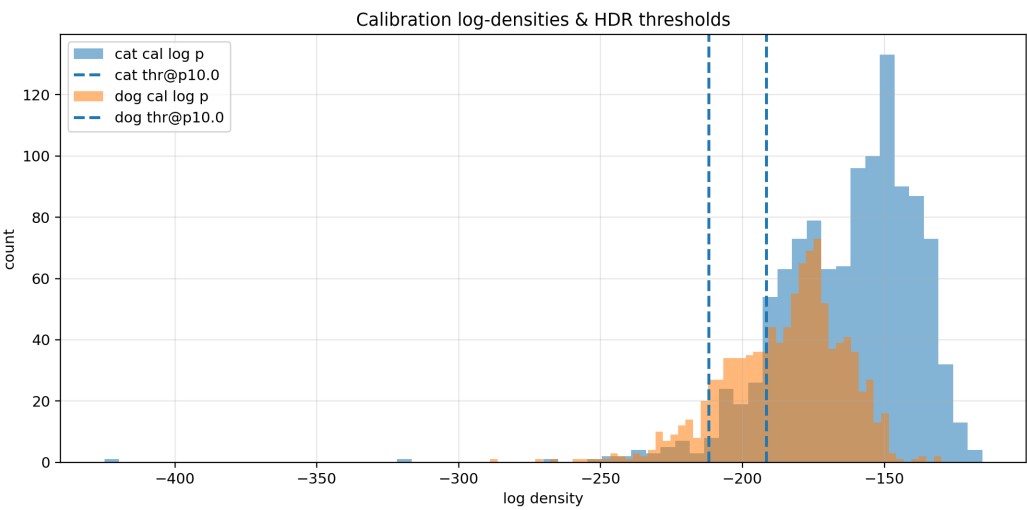

Figure 7: **An Example of HDR.** We shown an example of HDR for the class of cats and dogs. Dashed vertical lines mark the HDR thresholds at the $10\%$ quantile. Samples to the right of the threshold belong to the most probable $10\%$ of the calibration distribution for that class. Samples to the left of the threshold are deemed outside the HDR and treated as potential hallucinations.

**Highest Conditional Density Regions.** We emphasize a connection between Highest Conditional Density Regions and HDRs. Specifically, when the latent variable $Z$ only has *one* latent state, the $\delta$-hallucination in this special occasion is the expectation of the target distribution falling out of the HDRs of a certain mass that induces a density bound of $\delta$. We then extend this concept to the distributions correlated with a latent variable with *more than one* states. Namely, we introduce the concept of Highest Conditional Density Regions (HCDRs) and define it as follows.

**Definition B.1** (Highest Conditional Density Regions). Let $d$ be a distribution and $Z$ a latent variable correlated with $d$. Let $d_i$ denote the conditional probability of $d$ when knowing $Z = Z_i$, here $Z_i,\ i \in [N]$ is one of the $N$ states of $Z$. This explicitly writes as

$$d_i = d \mid \{Z = Z_i\}.$$

We define the Highest Conditional Density Regions $S_M$ as the smallest region on which the integral of $d_i$ is $M$.

Figure 8 shows the difference of HCDR and HDR.

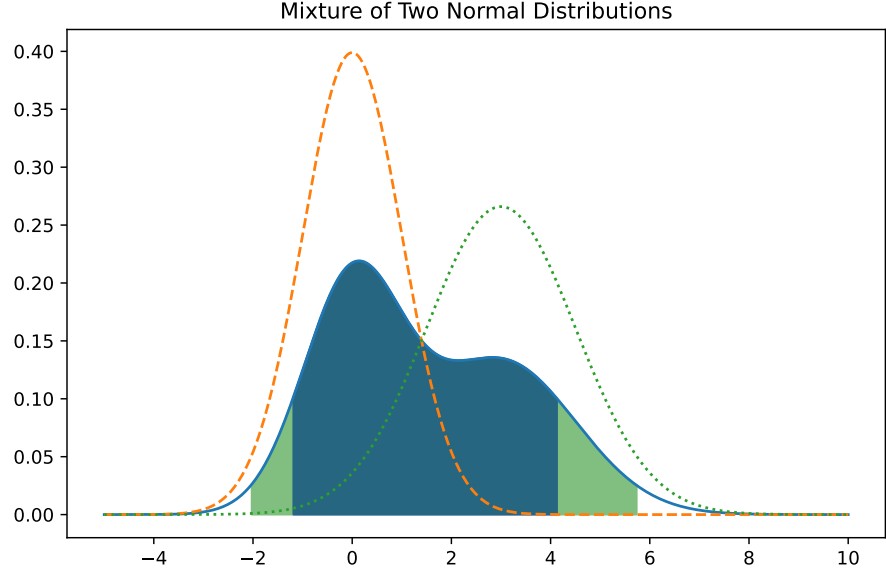

Figure 8: **An Example of HCDR vs. HDR**. We show the difference between HCDR and HDR for a mixture of two normal distributions. The blue region denotes HDR, whereas the green and blue region together denote HCDR. $\delta$ denotes the bound of the HDR (10%), and $\delta_1$ (5%) denotes the bound for the conditional probabilities. Though HDR is encapsulated in HCDR in this example, HDR might contain regions outside HCDR in other cases, meaning HCDR is not simply an expansion of HDR.

## C  PROOFS OF MAIN TEXT

### C.1  PROOF OF THEOREM 4.1

To prove the existence of $\delta$-hallucination, we state the following lemma.

**Lemma C.1.** The estimator $A^*(X)$ that minimizes the expected quadratic loss over $A(X)$ is

$$A^*(X) = \mathbb{E}_{A(X)}[A(X)].$$

*Proof.* As defined in Definition 2.1, for $A^*(X)$, the loss over $A(X)$ is

$$\ell_{A(X)}(A^*(X)) = \mathbb{E}_{A(X)}[\|A^*(X) - a\|_2^2]$$

$$= \int_{a \in \mathcal{A}} \|A^*(X) - a\|_2^2 \cdot f_{A(X)}(a)\mathrm{d}a, \tag{C.1}$$

where $\mathcal{A}$ is the output domain of $A(X)$ (the set of all possible outputs). By our notation defined in Section 2, $f_{A(X)}$ is the probability density function of $A(X)$.

Now, for an $A^*$ that minimizes the loss at $X$. We have its gradient at $A(X)$ to be $0_{d_a}$ ($d_a$ is the output dimension as in Definition 3.1).

$$\nabla \ell_{A(X)}(A^*(X)) = 0.$$

Combine the above equation with (C.1) we have

$$\nabla(\int_{a \in \mathcal{A}} \|A^*(X) - a\|_2^2 \cdot f_{A(X)}(a)\mathrm{d}a) = 0. \tag{C.2}$$

Since the $\nabla$ here denotes the gradient of $A^*(X)$, we have

$$\nabla(\int_{a \in \mathcal{A}} \|A^*(X) - a\|_2^2 \cdot f_{A(X)}(a)\mathrm{d}a)$$

$$= \int_{a \in \mathcal{A}} \nabla \|A^*(X) - a\|_2^2 \cdot f_{A(X)}(a)\mathrm{d}a$$

$$= \int_{a \in \mathcal{A}} \nabla(\|A^*(X)\|_2^2 - 2A^*(X)^\top a) \cdot f_{A(X)}(a)\mathrm{d}a \quad (\|A(X)\|_2^2 \text{ is erased when taking the gradient})$$

$$= \int_{a \in \mathcal{A}} (2A^*(X) - 2a) \cdot f_{A(X)}(a)\mathrm{d}a$$

$$= 2 \int_{a \in \mathcal{A}} A^*(X) \cdot f_{A(X)}(a)\mathrm{d}a - 2 \int_{a \in \mathcal{A}} A(X) \cdot f_{A(X)}(a)\mathrm{d}a$$

$$= 2A^*(X) - 2 \int_{a \in \mathcal{A}} a \cdot f_{A(X)}(a)\mathrm{d}a. \qquad\qquad (\text{By } \int_{\mathcal{A}} f_{A(X)}(a)da = 1)$$

Combine the above result with (C.2), we have

$$2A^*(X) - 2 \int_{a \in \mathcal{A}} A(X) \cdot f_{A(X)}(a)\mathrm{d}a = 0.$$

Thus $A^*$ is

$$A^*(X) = \int_{a \in \mathcal{A}} a \cdot f_{A(X)}(a)\mathrm{d}a = \mathbb{E}[A(X)]. \tag{C.3}$$

This completes the proof. $\qquad\qquad\qquad\qquad\qquad\qquad\qquad\qquad\qquad\qquad\qquad\qquad\square$

**Theorem C.1** (Existence of $\delta$-Hallucination under Single Input; Theorem 4.1 Restate)**.** For an input $X$, there exists infinitely many distributions of $A(X)$ and $Z$ such that for an estimator $A^*$ that minimizes the expected quadratic loss defined in Definition 2.1 over $A(X)$, it is bound to $\delta$-hallucinate at $X$.

*Proof.* By Lemma C.1, we have

$$A^*(X) = \mathbb{E}_{A(X)}[A(X)].$$

We now construct a wide range of distribution of $A(X)$ and $Z$ that satisfies

$$f(A^*(X); Z) \le \delta.$$

Let $N$ (number of latent states) be any positive number. Then, let $A(X; Z_i), i \in [N-1]$ be a normal distribution of the form

$$f_{A(X;Z_i)}(a) := (2\pi)^{\frac{-d_a}{2}} \det(\Sigma_i)^{-\frac{1}{2}} \exp\left(-\frac{1}{2}(X - \mu_i)^\top \Sigma_i^{-1}(X - \mu_i)\right).$$

By the requirements of normal distributions, $\Sigma_i$ are positive-definite matrices in $\mathbb{R}^{d_a \times d_a}$, and $\mu_i$ are $d_a$-dimensional vectors.

This is also denoted as

$$A(X; Z_i) \sim \mathcal{N}(\mu_i, \Sigma_i),$$

where $\mathcal{N}(\mu_i, \Sigma_i)$ denotes a normal distribution of mean $\mu_i$ and covariance matrix $\Sigma_i$ by convention.

Then, define $\mu_i$ to satisfy

$$f_{A(X;Z_i)}(0_{d_a}) = (2\pi)^{\frac{-d_a}{2}} \det(\Sigma_i)^{-\frac{1}{2}} \exp\left(-\frac{1}{2}\mu_i^\top \Sigma_i^{-1}\mu_i\right) \le \delta$$

For any $\delta > 0$, this $\mu_i$ always exists. We give the following example.

$$\mu_i = m_i 1_{d_a},$$

where $m_i$ is

$$\sqrt{\frac{-2\ln(\delta) - \ln(\det(\Sigma_i))}{1_{d_a}^\top \Sigma_i 1_{d_a}}}. \qquad\qquad (\delta \in (0,1])$$

The probability density is

$$f_{A(X;Z_i)}(0_{d_a}) = (2\pi)^{\frac{-d_a}{2}} \det(\Sigma_i)^{-\frac{1}{2}} \exp\left(-\frac{1}{2}\mu_i^\top \Sigma_i^{-1} \mu_i\right)$$

$$= (2\pi)^{\frac{-d_a}{2}} \det(\Sigma_i)^{-\frac{1}{2}} \exp\left(-\frac{1}{2}m_i^2 1_{d_a}^\top \Sigma_i 1_{d_a}\right)$$

$$= (2\pi)^{\frac{-d_a}{2}} \det(\Sigma_i)^{-\frac{1}{2}} \exp\left(-\frac{1}{2}\frac{-2\ln(\delta) - \ln(\det(\Sigma_i))}{1_{d_a}^\top \Sigma_i 1_{d_a}} \cdot 1_{d_a}^\top \Sigma_i 1_{d_a}\right)$$

$$= (2\pi)^{\frac{-d_a}{2}} \det(\Sigma_i)^{-\frac{1}{2}} \exp\left(\ln(\delta) + \frac{1}{2}\ln(\det(\Sigma_i))\right)$$

$$= (2\pi)^{\frac{-d_a}{2}} \det(\Sigma_i)^{-\frac{1}{2}} \cdot \det(\Sigma_i)^{\frac{1}{2}} \delta$$

$$= (2\pi)^{\frac{-d_a}{2}} \delta$$

$$\leq \delta. \qquad\qquad (C.4)$$

This means our definition of $\mu_i$ is valid.

For simplicity, let $p_i$ denote $\Pr[Z = Z_i]$:

$$p_i := \Pr[Z = Z_i].$$

Now, define $A(X; Z_N)$ to be

$$A(X; Z_N) \sim \mathcal{N}(-\sum_{i \in [N-1]} \frac{p_i}{p_n}\mu_i, \Sigma_N). \qquad\qquad (C.5)$$

Let $\mu_N$ denote $-\sum_{i \in [N-1]} p_i/p_n \cdot \mu_i$.

Let $m_N \in \mathbb{R}$ be

$$m_N := \delta^{-\frac{2}{d_a}}.$$

Then let $\Sigma_N$ be defined as

$$\Sigma_N := \frac{1}{m_N} \cdot I_{d_a},$$

which is positive definite.

This means

$$\Sigma_N^{-1} = m_N \cdot I_{d_a}$$

is also positive definite.

Thus we have

$$\exp\left(-\frac{1}{2}\mu_N^\top \Sigma_N^{-1} \mu_N\right) \leq \exp(0) = 1.$$

Then along with (C.5) we have

$$f_{A(X;Z_N)}(0_{d_a}) = (2\pi)^{\frac{-d_a}{2}} \det(\Sigma_N)^{-\frac{1}{2}} \exp\left(-\frac{1}{2}\mu_N^\top \Sigma_N^{-1} \mu_N\right)$$

$$\leq \det(\Sigma_N)^{-\frac{1}{2}}$$

$$= (m_N^{d_a})^{-\frac{1}{2}}$$
$$= \delta^{-\frac{2}{d_a} \cdot \frac{-d_a}{2}}$$
$$= \delta. \tag{C.6}$$

Recall in (C.3) we have proven $A^*$ to be the expectation of $A$. This means for the distribution $A(X)$ we've constructed here, we have

$$\begin{aligned}
A^*(X) &= \mathbb{E}[A(X)] \\
&= \mathbb{E}_Z[\mathbb{E}_A[A(X;Z)]] \\
&= \sum_{i=1}^{N} \Pr[Z = Z_i]\mathbb{E}[A(X;Z_i)] \\
&= \sum_{i=1}^{N} p_i \mu_i \\
&= \sum_{i=1}^{N-1} p_i \mu_i + p_N \mu_N \\
&= \sum_{i=1}^{N-1} p_i \mu_i + p_N(-\sum_{i=1}^{N-1} \frac{p_i}{p_N}\mu_i) \\
&= 0.
\end{aligned}$$

Combining the fact of $A^*(X) = 0$ with (C.4) and (C.6) satisfies the condition of $\delta$-hallucination defined in Definition 3.2. This completes the proof. $\square$

## C.2 PROOF OF COROLLARY 4.2.1

**Corollary C.1.1** (Existence of $\delta$-Hallucination under Multiple Inputs; Corollary 4.2.1 Restate). For a set of input $X_j, j \in [S]$, there exists infinitely many distributions of $A(X_j)$ and $Z$ such that any estimator minimizing the expected quadratic loss defined in Definition 2.1 is bound to $\delta$-hallucinate at $X$.

*Proof.* Construct every $A(x_j)$ according to the construction of Appendix C.1. This makes every $A^*(X_j), j \in [S]$ to fall out of the non-hallucinating region. This completes the proof. $\square$

## C.3 PROOF OF APPENDIX C.3

**Theorem C.2** (Existence of $\delta$-Hallucination on Semi-Optimal Estimators Under Single Input; Theorem 4.2 Restate). For an input $X$, there exists infinitely many distributions of $A(X)$ and $Z$ such that if an estimator $A'$ is within a distance of $\epsilon$ to the optimal estimator $A^*$, which writes as

$$\|A'(X) - A^*(X)\|_2 \le \epsilon,$$

then $A'(X)$ is bound to $\delta$-hallucinate.

*Proof.* By Lemma C.1, we have

$$A^*(X) = \mathbb{E}_{A(X)}[A(X)].$$

Thus we have

$$\|A'(X) - \mathbb{E}[A(X)]\|_2 \le \epsilon. \tag{C.7}$$

Let $N$ be any *even* number in $N^+$.

Construct

$$A(X; Z_i) \sim \mathcal{N}(\mu_i, I_{d_a}).$$

Let $\mathbb{E}[A(X)] = \sum_{i=1}^{N} p_i \mu_i$ be 0. Here $p_i = \Pr[Z = Z_i]$. Then by (C.7), we have

$$\|A'(X) - 0\|_2 \leq \epsilon.$$

Let $v_0$ denote $A'$. The probability of $v_0$ in $A(X; Z_i)$ is

$$(2\pi)^{\frac{-d_a}{2}} \exp\left(-\frac{1}{2}(v_0 - \mu_i)^\top (v_0 - \mu_i)\right) = (2\pi)^{\frac{-d_a}{2}} \exp\left(-\frac{1}{2}\|v_0 - \mu_i\|_2^2\right).$$

Set $\|\mu_i\|_2 \geq \sqrt{-2\ln\delta} + \epsilon$, we have

$$(2\pi)^{\frac{-d_a}{2}} \exp\left(-\frac{1}{2}\|v_0 - \mu_i\|_2^2\right) \leq \exp\left(-\frac{1}{2}\|v_0 - \mu_i\|_2^2\right)$$

$$\leq \exp\left(-\frac{1}{2}(\|v_0 - \mu_i\|_2 - \|v_0\|_2)^2\right)$$

$$\leq \exp\left(-\frac{1}{2}(\sqrt{-2\ln\delta} + \epsilon - \epsilon)^2\right)$$

$$\leq \delta.$$

Finally, let

$$\mu_i = -\frac{p_{N-i}}{p_i} \mu_{N-i}. \qquad\qquad (N \text{ has been set to be even})$$

This ensures $\sum_{i=1}^{N} p_i \mu_i$ to be 0.

The last constraint can coexist with $\|\mu_i\|_2 \geq \sqrt{-2\ln\delta} + \epsilon$ in infinitely many constructions of $\mu_i, i \in [N]$ (e.g., $\mu_i = C \cdot i(N-i)(\sqrt{-2\ln\delta} + \epsilon)/p_{N-i} \cdot 1_{d_a}$ for any $C > 1$). This completes the proof.

$\square$

## C.4 PROOF OF THEOREM 4.3

**Theorem C.3** (Existence of $\delta$-Hallucination at Tilted Input; Theorem 4.3 Restate)**.** Let $B_\delta$ denote the bound of all hints $\delta_i, i \in [N]$, defined as

$$B_\delta := \sup_{i \in [N]} \|\delta_i\|_2.$$

For an $L$-Lipschitz estimator $A^*$ satisfying Definition 2.2, there exists infinitely many distributions of $A(X; Z)$ such that $\delta$-Hallucination happens on all $X + \delta_i$. That is, $A^*(X + \delta_i)$ does not fall into the region where $f_{A(X; Z_i)} \geq \delta$ for any $i \in [N]$ by Definition 3.1.

*Proof.* Let $A(X; Z_i)$ be a normal distribution with a mean of $\mu_i$ and a covariance matrix of $\Sigma_i$. Construct $\sum_{i=1}^{N} p_i \mu_i = 0_{d_a}$, where $p_i = \Pr[Z = Z_i]$.

Because $A^*$ is $L$-Lipschitz, we have

$$\|A^*(X + \delta_i) - A^*(X)\|_2 \leq L\|X + \delta_i - X\|_2 = L\|\delta_i\|_2 \leq LB_\delta. \qquad (C.8)$$

See $LB_\delta$ as $\epsilon$, and $A^*(X + \delta_i)$ as different $A'$ in Theorem 4.2. Apply Theorem 4.2 to every $A(X + \delta_i)$. Thus, there are infinitely many distributions for $A^*(X + \delta_i)$ to $\delta$-hallucinate over $A(X)$. This completes the proof.

$\square$

## C.5  Proof of Theorem 5.1

To prove Theorem 5.1, we state the following definitions amd assumptions.

We begin with the definition of means and variances for the variables of interest.

**Definition C.1** (Means and Variances; Definition 5.1 Restate). Let $\{Z_i\}_{i\in[N]}$ denote the possible states of the latent variable $Z$, with probabilities $p_i := \Pr[Z = Z_i]$. For each $i \in [N]$, define the conditional mean

$$\mu_i := \mathbb{E}[A(X; Z_i)].$$

We regard $\mu_i$ as a realization of a random variable distributed according to $d_i^\mu$. Let $\mu_i^d := \mathbb{E}_{d_i^\mu}[\mu_i]$ and $\sigma_i^d := \mathrm{Var}_{d_i^\mu}[\mu_i]$ denote the mean and variance of this distribution, respectively. Let $d^\mu$ denote the joint distribution of $(\mu_1, \ldots, \mu_N)$. We write $\mu^d := \mathbb{E}_{d^\mu}[\mu_1, \ldots, \mu_N]$ for its mean vector and $\sigma^d := \mathbb{E}[\sum_{i=1}^N (\mu_i - \mu_i^d)^2]$ as sum of variance.

We then provide the following assumptions applied to $\mu_i$ and $d_i^\mu$ in Definition C.1. In particular, we assume that the conditional means align around a common value and that the joint distributions of these conditional means are mutually independent.

**Assumption C.1.** We impose the following conditions on the distributions defined in Definition C.1:
1. *Identical means*: There exists a constant $\mu_0 \in \mathbb{R}$ such that $\mu_i^d = \mu_0$, for all $i \in [N]$.
2. *Independence*: The distributions $\{d_i^\mu\}_{i=1}^N$ are mutually independent.

We now characterize hallucination events in terms of output regions that correspond to high $(> \delta)$ conditional probability under each latent state.

**Definition C.2** (High Conditional Density Regions; Definition 5.2 Restate). We define $U_i^\delta$ to be

$$U_i^\delta := \{a \mid f(a; Z_i) > \delta\},$$

which is the region with posterior probability of $Z = Z_i$ larger than $\delta$.

**Remark C.1** (Remark 5.2 Restate). By Definition C.2, $\delta$-hallucination of $A^*(X)$ is equivalent to

$$A^*(X) \notin U_i^\delta, \quad i \in [N].$$

We then define the following spheres covering $U_i^\delta$ in Definition C.2. Specifically, we enclose each $U_i^\delta$ within the smallest possible sphere centered at the corresponding mean $\mu_i$.

**Definition C.3** (Minimal Covering Spheres; Definition 5.3 Restate). For each $i \in [N]$, let $U_i^\delta \subset \mathbb{R}^{d_a}$ denote the $\delta$-high density region associated with state $Z_i$. Define $B_i^\delta(r)$ as the closed Euclidean ball of radius $r$ centered at $\mu_i$. The minimal covering radius is

$$r_i := \inf_{r_i \in \mathbb{R}^+} \{U_i^\delta \subset B_i^\delta(r_i)\}.$$

Thus $B_i^\delta(r_i)$ is the smallest sphere centered at $\mu_i$ that contains $U_i^\delta$. Finally, define the uniform covering radius

$$r = \max_{i\in[N]}\{r_i\}.$$

Next, we state the following axillary lemmas.

**Lemma C.2** (Paley-Zygmund Inequality). For any non-negative random variable $T$ and any $\theta \in [0,1]$, we have

$$\Pr[T > \theta \cdot \mathbb{E}[T]] \geq (1-\theta)^2 \frac{(\mathbb{E}[Z])^2}{\mathbb{E}[Z^2]}.$$

**Lemma C.3** (Chebyshev Inequality). For any random variable $T$, we have

$$\Pr[|T - \mathbb{E}[T]| \geq a] \leq \frac{\mathrm{Var}[T]}{a^2}, \quad \text{for all constant} \quad a,$$

where $\mathrm{Var}[T]$ is the variance of $T$.

**Lemma C.4** (Cauchy Inequality). For any $n \in \mathbb{N}^+$ along with two sets of variables $x_1, x_2, \cdots, x_n$ and $y_1, y_2, \cdots, y_n$, they satisfy

$$(\sum_{i=1}^n x_i y_i)^2 \leq (\sum_{i=1}^n x_i^2)(\sum_{i=1}^n y_i^2).$$

By Lemma C.3 and Lemma C.4, we derive a bound for the probability of distances between the loss minimizing estimator and the mean of $d^\mu$ defined in Definition C.1 which is $\mu_0$ by Assumption C.1 as follows.

**Lemma C.5** (Probability Upper Bound of Distance between $A^*(X)$ and $\mu_0$ in Assumption C.1). Let $A^*$ be the optimal estimator over $A$. Then for any $d_1 > 0$ we have

$$\Pr\big[\|\mu_0 - A^*(X)\|_2^2 \geq d_1^2\big] \leq \frac{(\sum_{i=1}^N p_i^2)\sigma^d}{d_1^2}.$$

*Proof.* By Lemma C.3, we have

$$\Pr\big[(A^*(X) - \mu_0)^2 \geq d_1^2\big] \leq \frac{\mathbb{E}[(A^*(X) - \mu_0)^2]}{d_1^2}$$

$$= \frac{\mathbb{E}[(\sum_{i=1}^N p_i \mu_i - \mu_0)^2]}{d_1^2}$$

$$= \frac{\mathbb{E}[[\sum_{i=1}^N p_i(\mu_i - \mu_0)]^2]}{d_1^2}$$

$$\leq \frac{\mathbb{E}[(\sum_{i=1}^N p_i^2)[\sum_{i=1}^N (\mu_i - \mu_0)^2]]}{d_1^2} \qquad \text{(By Lemma C.4)}$$

$$= \frac{(\sum_{i=1}^N p_i^2)\mathbb{E}[\sum_{i=1}^N (\mu_i - \mu_0)^2]}{d_1^2}$$

$$= \frac{(\sum_{i=1}^N p_i^2)\sigma^d}{d_1^2}.$$

This completes the proof. □

In addition, by Lemma C.2, we derive a lower bound of the probability of distances between $\mu_i$ defined in Definition C.1 and $\mu_0$ defined in Assumption C.1.

**Lemma C.6** (Lower Bound on the Probability of Distance between $\mu_i$ in Definition C.1 and $\mu_0$ in Assumption C.1). For $i \in [N]$, let $\mu_i$ and $\mu_0$ be as defined in Definition C.1 and Assumption C.1. We have, for any $\theta \in [0, 1]$,

$$\Pr\big[\|\mu_i - \mu_0\|_2^2 \geq \theta\sigma_i^d\big] \geq (1 - \theta)^2 K_i^\mu.$$

*Proof.* Because $\|\mu_i - \mu_0\|_2^2 \geq 0$, by Lemma C.2, set $T$ in Lemma C.2 to be $\|\mu_i - \mu_0\|_2^2$, and we have

$$\Pr\big[\|\mu_i - \mu_0\|_2^2 \geq \theta\mathbb{E}[\|\mu_i - \mu_0\|_2^2]\big] \geq (1 - \theta)^2 \frac{\mathbb{E}[(\mu_i - \mu_0)^2]^2}{\mathbb{E}[\|\mu_i - \mu_0\|_2^4]} = (1 - \theta)^2 K_i^\mu.$$

Combining with

$$\mathbb{E}[\|\mu_i - \mu_0\|_2^2] = \sigma_i^d,$$

we have

$$\Pr\big[\|\mu_i - \mu_0\|_2^2 \geq \theta\sigma_i^d\big] \geq (1-\theta)^2 K_i^\mu.$$

This completes the proof. □

Therefore, by Lemma C.5 and Lemma C.6, combined with Definition C.3, we prove the lower bound of the probability of hallucination.

**Theorem C.4** (Hallucination Probability Lower Bound; Theorem 5.1 Restate). Let $(A(X), Z)$ satisfy Assumption 5.1. For each $i \in [N]$, let $\mu_i, \sigma_i^d$ be as in Definition 5.1, let $\mu_0$ be as in Assumption 5.1, and let $r_x$ be as in Definition 5.3. Define

$$d := (\sum_{j=1}^N p_j^2 \sigma_j^d)^{1/2}, \quad \theta_i(\alpha) := \frac{(\alpha d + r_x)^2}{\sigma_i^d}, \quad \alpha > 1, \quad \text{and} \quad K_i^\mu := \frac{(\mathbb{E}[(\mu_i - \mu_0)^2])^2}{\mathbb{E}[(\mu_i - \mu_0)^4]}.$$

If for every $i \in [N]$ there exists $\alpha_i > 1$ such that $\theta_i(\alpha_i) \leq 1$, then

$$P_H^\delta > \prod_{i=1}^N (P_i K_i^\mu),$$

where $P_H^\delta$ denotes the probability that the optimal estimator $A^*$ $\delta$-hallucinates at $X$ (equivalently, $A^*(X) \notin U_i^\delta$ for all $i \in [N]$, with $U_i^\delta$ as in Definition 5.3).

*Proof.* By Lemma C.5, for every $i \in [N]$, we have

$$\Pr\big[\|\mu_0 - A^*(X)\|_2^2 \geq d_i^2\big] \leq \frac{(\sum_{i=1}^N p_i^2)\sigma^d}{d_i^2}.$$

This means

$$\Pr\big[\|\mu_0 - A^*(X)\|_2^2 \leq d_1^2\big] \geq 1 - \frac{(\sum_{i=1}^N p_i^2)\sigma^d}{d_1^2}. \tag{C.9}$$

By Lemma C.6, we have, for every $i \in [n]$

$$\Pr\big[\|\mu_i - \mu_0\|_2^2 \geq \theta_i\sigma_i^d\big] \geq (1-\theta_i)^2 K_i^\mu. \tag{C.10}$$

Then, Definition C.3, the probability for $A^*$ to fall out of the region with a conditioned probability of $A(X; Z_i)$ no less than $\delta$ is at least

$$\begin{aligned}
\Pr\big[A^*(X) \notin U_i^\delta\big] &\geq \Pr\big[A^*(X) \notin B_i^\delta(r_i)\big] \\
&\geq \Pr[\|A^*(X) - \mu_0\|_2 \leq d_i] \cdot \Pr[\|\mu_i - \mu_0\|_2 \geq d_i + r_x] \\
&\geq (1 - \frac{(\sum_{i=1}^N p_i^2)\sigma^d}{d_i^2})((1-\theta_i)^2 K_i^\mu) \qquad \text{(By (C.9) and (C.10))} \\
&= (1 - \frac{1}{\alpha_i^2})(1-\theta_i)^2 K_i^\mu.
\end{aligned}$$

Set $\alpha_i$ to maximize

$$(1 - \frac{1}{\alpha_i^2})(1-\theta_i)^2,$$

which is equivalent to maximizing $P_i$.

Then we have

$$\Pr\big[A^*(X) \notin U_i^\delta\big] \geq P_i K_i^\mu.$$

Given $d_i^\mu, i \in [N]$ are independent to each other, we have

$$\Pr\big[A^*(X) \notin U_i^\delta, i \in [N]\big] \geq \prod_{i=1}^N P_i K_i^\mu.$$

The left-hand side is equivalent to $P_h^\delta$ (see Definition C.2 and Remark C.1).

This completes the proof. □

## D    DERIVATION TO CROSS-ENTROPY LOSS

In this section, we derive the cross-entropy loss version of our results in Section 4.

**Definition D.1** (Cross-Entropy Loss). For an input $X$ and an according possible output $a \in \mathcal{A}$, given a target probability density $q_X^a \in [0,1]^C$ and a model-estimated distribution $p_X \in [0,1]^C$ over $C$ classes, let $q_X^a(t)$ and $p_X(t)$ denote their $t$-th entry respectively. The cross-entropy loss at $X$ is defined as

$$\mathcal{L}(q_X^a, p_X) = -\sum_{t \in [C]} q_X^a(t) \log p_X(t),$$

where $q_X^a(t) \geq 0$, $\sum_{t \in [C]} q_X^a(t) = 1$, $p_X(t) \geq 0$, and $\sum_{t \in [C]} p_X(t) = 1$.
We define the total loss at $X$ as the expectation of loss over $\mathcal{A}$ at all $a$, that is

$$E_a(\mathcal{L}(q_X^a, p_X)).$$

Comparing to the notation in Section 4, the predictor $A^*$ at input $X$ outputs the predicted probabilities $A^*(X)$, which can be noted here as

$$[A^*(X)](t) := p_X(t), t \in [C],$$

We now prove the existence of $\delta$-hallucination under cross-entropy loss.

**Theorem D.1** (Existence of $\delta$-Hallucination under Cross-Entropy Loss). For an input $X$, there exists infinitely many target distributions $A(X)$ such that the $A^*$ minimizing the cross-entropy loss defined in Definition D.1 at $X$ $\delta$-hallucinates.

*Proof.* We first calculate the loss minimizing $A^*$ at $X$.

$$E_a(\mathcal{L}(q_X^a, p_X))$$
$$= \int_{\mathcal{A}} p(a)[-\sum_{t \in [C]} q_X^a(t) \log p_X(t)]da$$
$$= \sum_{t \in [C]} (-\log p_X(t))[\int_{\mathcal{A}} q_X^a(t)da]$$
$$= \sum_{t \in [C]} (-\log p_X(t)) E_a q_X^a(t).$$

Thus by Gibbs Inequality, we have the loss minimizing $p_X(t)$ of $E_a(\mathcal{L}(q_X^a, p_X))$ is

$$p_X(t) = E_a q_X^a(t), t \in [C].$$

We then construct the latents that induce the $\delta$-hallucination at $X$.

Define the probability distribution under each $Z_i$ as

$$A(q_X^a | Z = Z_i) \sim \mathcal{N}(q_i, d), \ i \in [N],$$

in which $q_i$ is

$$q_i(t) := e_t^{(C)},$$

and

$$d \leq -\frac{N-1}{N\ln(\delta^2)}.$$

Then let $P(Z_i) = 1/N$, we have $p_X$ equals

$$p_X := \frac{\sum_{i=1}^{N} e_i^{(C)}}{N}.$$

Then

$$P(p_x|Z = Z_i)$$
$$= \frac{1}{\sqrt{2\pi d}} \exp\left(-\frac{(p_X - q_i)^2}{2d}\right)$$
$$= \frac{1}{\sqrt{2\pi d}} \exp\left(-\frac{N-1}{2dN}\right)$$
$$\leq \frac{1}{\sqrt{-2\pi \frac{N-1}{N\ln(\delta^2)}}} \exp\left(-\frac{N-1}{-2\frac{N-1}{N\ln(\delta^2)}N}\right)$$
$$\leq \frac{1}{\sqrt{-\pi \frac{1}{\ln(\delta^2)}}} \frac{\delta^2}{2}$$
$$\leq \frac{\delta^2 \ln(\delta^{-1})}{\sqrt{2\pi}}$$
$$\leq \frac{\delta^2(\delta^{-1} - 1)}{\sqrt{2\pi}}$$
$$\leq \delta,$$

for every $i$.

This completes the proof.

$\square$