# OpenReview forum: "Are Hallucinations Bad Estimations?"
_ICLR.cc/2026/Conference — Submitted to ICLR 2026_

### Official Review · Reviewer_2b9S · 2025-10-22

**Soundness:** 1
**Presentation:** 2
**Contribution:** 2
**Rating:** 2
**Confidence:** 4

**Summary:**

The paper aims to understand  hallucination as a statistical estimation problem. It assumes a latent variable $Z$ (the hidden truth) and an observed variable $X$ (the context or prompt). A ground-truth oracle $A(X)$ produces the correct output, leading to a joint distribution $p(Z, X, A(X))$. Under a squared loss $L(a, A(X)) = \|a - A(X)\|^2$, the Bayes-optimal estimator is defined as ${A^\*}(X) = \mathbb{E}[A(X)\mid X]$. The estimator $A^\*(X)$ is said to $\delta$-hallucinate at $X$ if, for every latent state $Z_i$, the conditional probability of its output satisfies $p(A^*(X)\mid Z_i)\le\delta$, where $p(a\mid Z_i)$ denotes the conditional distribution of the ground-truth oracle $A(X)$ given the hidden state $Z_i$.

The paper then provides several theorems (Theorems 4.1, 4.2, 4.3, and 5.1) demonstrating the existence of configurations of $(Z, X, A(X))$ such that the Bayes-optimal estimator $A^*$ is $\delta$-hallucinating. The paper further conducts a series of experiments on synthetic regression, open-ended QA, and text-to-image generation to illustrate that hallucination can persist even as the training loss decreases.

Intuitively, the paper attributes hallucination to the overlap of the conditional outcome distributions associated with different hidden states. When multiple latent states $Z_i$ are consistent with the same observation $X$, the Bayes-optimal estimator aggregates them (e.g., by taking the conditional mean), producing an output that lies in a low-probability region of all $p(a\mid Z_i)$.

**Strengths:**

1. The paper introduces an interesting perspective by framing hallucination as a consequence of aggregating conditional distributions over unseen or latent states. This conceptualization appears to be a novel contribution.

2. The work may inspire practitioners to develop more resilient modeling approaches, such as choosing more appropriate loss functions or designing datasets that reduce multimodal ambiguity, to mitigate hallucination, although these ideas are not fully developed in the paper.

**Weaknesses:**

1. I have a major concern with the proof of Theorem 4.1 (and, by extension, potentially the subsequent theorems). In the proof, the authors explicitly construct the conditional distributions $p(A(X)\mid Z_i)$ in an arbitrary manner. This is problematic because, according to the main text, $A(X)$ is defined as a function of $X$ only. Consequently, once $X$ is fixed, the distribution of $A(X)$ should also be fixed and cannot vary across different hidden states $Z_i$ (i.e., the influence of $Z_i$ on $A(X)$ should only through $X$). However, in the proof, the authors implicitly allow the distribution of $A(X)$ to depend on both $X$ and $Z$, which contradicts their earlier definition. Under this relaxed assumption, the theorem becomes quite trivial: it merely states that one can always construct several conditional distributions whose average (the Bayes-optimal mean) lies in a region of low probability under each of them. In other words, the result reduces to the obvious observation that the mean of disjoint modes can fall between them. This conceptual inconsistency substantially reduces the appeal of Theorem 4.1 and, in my view, is a critical flaw that justifies a rejection.

2. The experimental setups appear to be chosen arbitrarily and only demonstrate that minimizing the training loss does not necessarily eliminate hallucination. However, they do not provide evidence for the paper’s central theoretical claim, that hallucination arises from the aggregation of multiple latent states. The experiments neither instantiate the formal definition of $\delta$-hallucination nor directly measure the overlap between conditional distributions associated with different hidden states. As a result, the empirical section does not convincingly validate the proposed theory.

3. The paper claims novelty by attributing hallucination to a “structural phenomenon of estimation itself,” but it misses several recent works with a similar spirit that connect hallucination to the fundamental statistical limits of learning, such as [1] and [2]. The authors should at least discuss how these papers relate to the present work.

4. There are many typos:
   - In line 046, it is unclear what the expectation is taken over.
   - In Definition 2.1, the expression for $A^*$ does not render properly.
   - Definition 3.2 is ambiguous: it is unclear how a function $f$ can be evaluated on an identity, i.e., what $f(a = b)$ is intended to mean.

[1] Wu, Grama, and Szpankowski. *No Free Lunch: Fundamental Limits of Learning Non-Hallucinating Generative Models.* ICLR 2025.

[2] Karbasi, Montasser, Sous, and Velegkas. *(Im)possibility of Automated Hallucination Detection in Large Language Models.* COLM 2025.

**Questions:**

Please address the points in the Weaknesses section.

---

### Official Review · Reviewer_1TmN · 2025-10-27

**Soundness:** 2
**Presentation:** 3
**Contribution:** 2
**Rating:** 4
**Confidence:** 3

**Summary:**

This paper introduces a theoretical framework arguing that hallucinations can arise even in optimal estimators minimizing standard training losses. The authors define the concept of “δ-hallucination,” prove its existence under mild assumptions, and provide synthetic and real-data experiments showing that hallucinations persist even when training loss is minimized.

**Strengths:**

This paper offers a fresh statistical interpretation of hallucinations as inherent estimation artifacts. Notably, formal results are presented with theoretical proofs of the inevitability of hallucinations under common loss objectives. Experiments on synthetic and real data are presented to support the claim.

**Weaknesses:**

(a) My major concern is the disconnection between the experiment design on LLM and the theoretical claim. Experiments on LLM appear to show that training under the current paradigm increases the “resemble score” to common mistakes. How does the model accuracy behave during training? As model accuracy typically improves after fine-tuning on a specific dataset, it is unclear whether the observed increase in “resemble score” reflects a genuine theoretical phenomenon or simply results from overfitting or distributional mismatch between training and test data. It is also unclear whether this observation is made on the training or test split.

(b) Another disconnection is that the theoretical analysis focuses on losses defined over scalar variables, whereas practical applications to LLMs involve vector- or sequence-valued losses and estimators that are clearly not i.i.d. The former formulation does not appear to naturally extend to the latter.

**Questions:**

see weakness

---

### Official Review · Reviewer_XzXo · 2025-10-30

**Soundness:** 1
**Presentation:** 2
**Contribution:** 2
**Rating:** 2
**Confidence:** 4

**Summary:**

Overall, the paper is well-structured, and the authors’ attention to the conditional density of generated samples in the context of hallucination is promising. However, this paper is challenging to follow, presents work that is loosely related to current literature, and makes many broad claims on hallucinations that (1) have assumptions that do not clearly fit into the context of generative models and (2) are not supported by experiments. I do not believe this paper meets the high standards of ICLR, but I may be willing to adjust my score if my main concerns are addressed.



Summary

The authors formalize the notion of hallucinations of generative models and link hallucinations to a misalignment between loss minimization and producing “human-acceptable” outputs. The authors state that the crux of the misalignment comes from estimation errors due to miscalibration. For example, the Bayes-optimal predictor of a target distribution is the expectation of the targets given some input, rather than a sample of a single plausible input.

Hallucination is formalized as delta-hallucination: an output that lies outside a delta-neighborhood of every plausible outcome. Using this, they prove that loss-minimizing optimal estimators still delta-hallucinate and that a lower bound exists on delta-hallucination depending on the distribution.

The authors introduce the latent variable Z to account for multi-modality in A(X). Their key insight is that good estimate should align with a plausible latent variable rather than all. While it isn’t clearly explained, this appears to emphasize that outputs should be strongly associated with latents, or modes in this case.

The authors conduct a series of 3 experiments which are designed to show that minimizing loss functions does not prevent hallucinations.

For the first experiment, they construct a coin-flipping problem where a transformer is trained to predict the expected number of heads from a set of coins with varying head probability. Their experiment shows the straightforward result that the expectation of a random variable does not necessarily coincide in a high-density region of the random variable. It is not clear how this relates to hallucinations of generative models, since the outputs of generative models are stochastic and typically produce plausible (high density) samples.
The second experiment involves finetuning Qwen models with LoRA on a question-answer pair dataset (generated from other LLMs) and measuring the similarity of the finetuned models’ responses when given TruthfulQA questions to common misconceptions. The authors report that there are small increases in similarity between the responses and the misconceptions. Without knowing the details of the finetuning dataset (created by the authors), it is not clear why that is the case. It is also not clear how the results show that loss minimization does not eliminate delta-hallucination, as the model is trained on a dataset which is unrelated to the testing set (why would this reduce hallucinations on the unrelated test set in the first place?). Moreover, delta-hallucination is not defined or measured in this context.

The third experiment involves finetuning a SDv1.5 U-Net on a subset of the Animal Faces-HQ dataset. A datum is considered a hallucination (not a cat or a dog) if the probability of its dimension-reduced CLIP embedding does not surpass a log-density threshold. This has been set reasonably via a GMM in the reduced CLIP space. However, critical details of the experiment are missing. For example, it is not clear if conditioning prompts such as “a photo of a cat” are included in training of cat images, as this would help prevent drift of the UNet representations from those of the encoder. If this was not provided, the finetuning would make the representations of the UNet and text encoder far less compatible, leading to test-time hallucinations (as measured by their experiment). Moreover, there is no comment on if classifier-free-guidance is used at test time. It is well-known that this is necessary in order to produce coherent images from the SDv1.5 U-Net. If this was not done, the produced images would be incomprehensible, leading to hallucinations as measured by their method.

There are no experiments regarding the bound presented in the paper.

**Strengths:**

The work is well-structured, and the attention to plausibility of an output or explanation is a promising direction. This work has originality in considering hallucination as a product of the gap between model training and human criteria.

The remarks are helpful in clarifying the intuition behind the definitions and theorems

The GMM method in reduced CLIP space in experiment 3 is innovative. I believe it is an appropriate way to measure delta-hallucinations as defined by your method. I am interested in learning more about that.

**Weaknesses:**

One could argue that the stochastic nature of generative models (LLMs, Diffusion models) addresses the structural misalignment of A*(X). For example, diffusion and rectified flow models can create many unique images given the same conditional input (e.g., text). LLMs can sample many new completions from the same prefix. Each of these may be plausible and do not match some expectation A*(X). Hence, if generative models do not fit the assumptions of the structural misalignment, then the authors' subsequent derivations might not necessarily apply to generative models.

The statement “we represent the output of the model as a probability distribution”(line 117)  is not consistent with the definitions (e.g., 2.1) presented in the paper. It is not clear how the sampling (output) of generative models connects to the assumptions made in the paper.

Figure 2 is not clear – the connection with conditional probability given the latent variable should be emphasized in the figure.
Definition 3.2 is not clear in that f() appears to be the probability that a latent variable generates a label that is consistent with the estimator. In other words, it is not clear that it is the conditional density evaluated on A*(X). Why not just use the expression in the key insight in this inequality? If the max is less than delta, then you have a delta-hallucination.

The experiments are missing details for reproducibility. Are each of the coins flipped M times to construct the coin-flipping dataset, or is it M total times? What were the heads probabilities for the coins, and how were they chosen?

Claims on existence of hallucinations on “Tilted Input” and on bounds of the hallucination rate of any optimal estimator are not mentioned again in the experiments or verified.

Minor:

All mathematical expressions, especially those outside of theorems/definitions, should have numerical identifiers on the right.

Definition 2.1 appears to have typos for A*(X)

**Questions:**

How does the estimator from the initial definitions connect to generative models, which are stochastic?

What is a hint delta_i in line 33, and does this constitute in practice? Is this side information that conditions the models outputs?

What are the missing experimental details? (see summary and weaknesses)

Can you design an experiment to verify the hallucination bound stated in this work?

---

### Official Review · Reviewer_mL1P · 2025-11-01

**Soundness:** 2
**Presentation:** 3
**Contribution:** 2
**Rating:** 6
**Confidence:** 3

**Summary:**

The paper reframes hallucination in generative models as a structural misalignment between loss-minimizing objectives and human expectations, introducing the concept of δ-hallucination to formalize outputs that fail to match any plausible mode. It shows that even Bayes-optimal estimators cannot eliminate hallucinations, establishing theoretical lower bounds on their persistence. Empirical validation across synthetic and real-world tasks supports the claim that hallucination are inevitable when single point estimates are considered rather than the full distribution.

**Strengths:**

* The paper provides a formal analysis of why hallucinations are bound to occur in generative models.
* The paper provides a high probability lower bound on hallucinate rate for generic data distributions.
* The paper includes experiments on coin aggregation, TruthfulQA, and text-to-image to validate the claims.
* The paper is easy to understand and well written.

**Weaknesses:**

* The main claim is not correct L48-50: If the true conditional distribution Pr[A(X)] = Pr[A(x) | X = x] is multimodel, then A⋆(X) average across all those possible outcomes and may fall in a low-probability region. This is only true for the Quadratic Loss in Definition 2.1. As described in  L103-105: "cross-entropy is a proper scoring rule: it's Bayes-optimal solution is the true conditional distribution P(Y |X)". So A⋆(X) will not average across all possible outcomes in case of the cross-entropy loss.

* Prior work already points out this cause for hallucinations: "Enhancing Hallucination Detection through Noise Injection, arXiv Feb 2025" (Sections 1 and 2).

* In Section 6.2, the evaluation is performed only on TruthfulQA. The paper does not discuss if the same conclusions can be reached for models with chain of thought reasoning.

* The paper only states that hallucinations are inevitable but does not propose or evaluate concrete mechanisms for reducing δ-hallucination, leaving its practical implications unclear.

**Questions:**

* The paper should explain the claim in L48-50 in more detail.
* Additional experiments on datasets such as GSM8k which require chain of thought reasoning will add value to the paper.
* The paper should include a discussion of prior work: "Enhancing Hallucination Detection through Noise Injection, arXiv Feb 2025".

---

### Meta-Review · Area_Chair_WCBn · 2026-01-07

**Summary:**

This paper presents a formal treatment of hallucinations as a mismatch between training objective and human acceptability. Authors present three different experimental settings combining real-world and synthetic tasks to validate their hypothesis.

**Reviewer Concerns:**

**Addressed**
None


**Unaddressed**
1. Problems with formulation: Ml1p also found issues with the main formulation, though that was not enough grounds for rejection. Xzxo found details in the method which were not clarified making it difficult to determine whether the treatment is justified. 1tmn pointed out an important problem: the treatment focuses on scalars while all language model variables are vectors.
2. 1tmn and 2b9s found there to be a discrepancy between the theoretical claim and the experimental design.
3. Reviewers found the paper challenging to read.
4. Relevant prior work from 2025 is not addressed. (ml1p, 2b9s)

**Reviewer Scores:**

There was no author response, so none of the reviewers got a chance to change their scores.

---

### Decision · Program_Chairs · 2026-01-26

Reject